# Coupling of polymerase-nucleoprotein-RNA in an influenza virus mini ribonucleoprotein complex

Huiling Kang[1,2,3,4,13], Yunxiang Yang[2,13], Yixiao Liu[2,13], Mingyu Li[2,13], Lejin Zhang [5,13], Yuqi Lin[5], Leander Witte [6], Kuang-Yu Chen [6], Wenya Song[7], Zhili Xu[5], Xiaojing He[8], Luke W. Guddat [9], Yu Guo [10], Liming Yan [2], Yan Gao [1], Ervin Fodor [6] ✉, Zihe Rao [1,2,3,5,10,11] ✉ & Zhiyong Lou [2,12] ✉

Influenza virus ribonucleoprotein complexes (RNPs), composed of the polymerase complex (FluPol), nucleoprotein (NP), and RNA, are essential for replication and transcription. We report atomic-resolution cryo-EM structures of mini-vRNPs in two states: FluPol located inside (State-In) or at the outer rim (State-Out) of the NP–RNA ring. In both states, the 5′ and 3′ termini of vRNA are bound to FluPol as previously reported. One NP (NP-0) contacts PA/PB1 of FluPol and binds the distal double-stranded vRNA promoter, with its D72–K90 loop inserting into the RNA fork; separated strands occupy NP-0 RNA-binding grooves. Grooves from other NPs form a continuous RNA-protective path, consistent with negative-strand RNA virus mechanisms. In State-In, interfaces for FluPol dimerization or Pol II interaction are blocked, but fully exposed in State-Out. These structures reveal detailed FluPol–NP–RNA coupling and suggest a conformational shift in RNPs during the viral life cycle.

Influenza A virus causes annual epidemics that result in highly contagious respiratory infections[1]. Influenza virus has a single-stranded negative-sense RNA genome with eight segments that typically encodes 11 or 12 viral proteins[2]. In the life cycle of the influenza virus, the viral genome RNA (vRNA) or the positive-sense complementary RNA (cRNA) segments are encapsidated by a virus-encoded nucleoprotein (NP), with their 5′ and 3′ termini clamped by the polymerase complex (FluPol; composed of PA, PB1, and PB2), thus forming the ribonucleoprotein complexes (RNPs)[2]. RNPs play essential roles in influenza virus replication and transcription, but the replication of full-

length vRNA or cRNA from cRNA-containing RNP (cRNP) or vRNA-containing RNP (vRNP) contrasts strikingly with the process of transcription[2,3]. Replication requires the oligomerization of FluPol, being supported by the branched RNP-like structures observed in cells and the oligomeric FluPol structures[4–12]. To transcribe viral mRNAs, the PA of FluPol associates with the C-terminal domain (CTD) of cellular RNA polymerase II (Pol II), which enables it to take 5′-capped primers from nascent Pol II transcripts[13,14].

The primary understanding of the architecture of the influenza virus RNP was obtained from cryo-electron microscopy (cryo-EM)

[1]Shanghai Institute for Advanced Immunochemical Studies and School of Life Science and Technology, ShanghaiTech University, Shanghai, China. [2]MOE Key Laboratory of Protein Science, School of Medicine, Tsinghua University, Beijing, China. [3]Guangzhou National Laboratory, Guangzhou, China. [4]Department of Clinical Laboratory, Xuanwu Hospital, National Clinical Research Center for Geriatric Diseases, Capital Medical University, Beijing, China. [5]Division of Life Sciences and Medicine, University of Science and Technology of China, Hefei, China. [6]Sir William Dunn School of Pathology, University of Oxford, Oxford, UK. [7]Department of Medical Oncology, The Fourth Hospital of Hebei Medical University, Shijiazhuang, Hebei, China. [8]MOE Key Laboratory of Molecular Biophysics, College of Life Science and Technology, Huazhong University of Science and Technology, Wuhan, China. [9]School of Chemistry and Molecular Biosciences, The University of Queensland, Brisbane, QLD, Australia. [10]State Key Laboratory of Medicinal Chemical Biology, College of Life Sciences and College of Pharmacy, Nankai University, Tianjin, China. [11]National Laboratory of Biomacromolecules, CAS Center for Excellence in Biomacromolecules, Institute of Biophysics, Chinese Academy of Sciences, Beijing, China. [12]State Key Laboratory of Virology, Wuhan, China. [13]These authors contributed equally: Huiling Kang, Yunxiang Yang, Yixiao Liu, Mingyu Li, Lejin Zhang. ✉e-mail: ervin.fodor@path.ox.ac.uk; raozh@tsinghua.edu.cn; louzy@mail.tsinghua.edu.cn

studies of a circular mini-RNP reconstituted in transfected cells that presents the minimal architecture of RNP, while the model was improved by cryo-EM studies on the filamentous RNPs extracted from virion or reconstituted in transfected cells[7,15–18]. In the models, two NPs contact FluPol to bind the single-stranded RNA downstream of the double-stranded distal part of the RNA promoter that projects away from the main body of FluPol, while the internal regions of RNPs comprise a twisted, antiparallel double helix of NP-RNA complexes. However, the resolution of these cryo-EM models is limited to the nano or subnano level, and the detailed knowledge of the coupling of FluPol, NP, and RNA in RNPs remains unclear.

In this work, we reconstitute the mini-vRNP by transfecting cells with plasmids that encode the vRNP components (i.e., pCAG-PA, pCAG-PB1, pCAG-PB2, and pCAG-NP that contain the genes encoding PA, PB1, PB2, and NP protein, respectively, as well as pPol I-vRNA that contains 3′ and 5′ promoters and a shortened internal region of vRNA), and determine the cryo-EM structures of the mini-vRNPs at atomic resolution. The structural information presented here furthers our understanding of the formation of the influenza virus RNP and explains its mechanism of replication and transcription.

## Results

### The reconstituted mini-vRNPs are functional

To verify whether the mini-vRNPs reconstituted in cells are functional, we inserted the gene encoding enhanced green fluorescent protein (eGFP) together with the 5′ and 3′ conserved termini of the sequence from the NS segment in the plasmid (pPol I-eGFP) to transcribe RNA for RNP reconstitution (Supplementary Fig. 1a). The fluorescence signal for eGFP was observed in cells transfected with the plasmids that encode all vRNP components after 24 h post-transfection and these signals peaked at 72 h post-transfection (Supplementary Fig. 1b). In contrast, no eGFP fluorescence was detected in the cells transfected with the plasmid that transcribes only RNA (Supplementary Fig. 1b). Together these data demonstrate that the gene encoding eGFP is transcribed, and the polymerase and NP in the mini-vRNPs formed in cells are functional.

### The mini-vRNPs present distinct architectures

The mini-vRNPs were purified by affinity chromatography followed by a glycerol gradient centrifugation (Supplementary Fig. 1c). The RNAs in the purified mini-RNPs were quantified by qRT-PCR, and the result shows a 1000-fold excess of vRNA over cRNA (Supplementary Fig. 2), confirming that the purified mini-RNPs are mainly mini-vRNPs. The fractions that were Western blot positive for both FluPol and NP were collected from gradient centrifugation and exhibited three distinct forms in negative-stain EM images (Supplementary Fig. 1d). In one form, NP molecules form a ring-shaped architecture and a bulk of density that may belong to FluPol filling the central hollow of the NP ring. We have defined this form as "State-In." In sharp contrast, the bulk density potentially corresponding to FluPol is attached at the outer rim of the NP ring in another form (defined as "State-Out"), which is similar to the previously reported mini-RNP[15,16]. In addition to these, we also observed some rings that were empty.

The samples were then analyzed by cryo-EM (Supplementary Figs. 3 and 4, and Supplementary Table 1). The 2D classifications of cryo-EM particles confirm the existence of State-In and State-Out particles, as well as the empty ring. The bulk densities present clear features of FluPol. In both states, NP rings consisting of seven, eight, nine, or ten NPs were observed, while the particles composed of eight NPs generated the best nominal resolution. Therefore, these particles (Supplementary Figs. 3b and 4b) were used for further 3D reconstructions (Supplementary Figs. 3c, d and 4c). The initial reconstructions of 265,190 particles for State-In mini-vRNP and 112,185 particles for State-Out mini-vRNP were obtained at 2.89-Å and 5.54-Å resolution, respectively. The relative flexibility of the RNP components limits

further improvement of the quality of the cryo-EM densities. Focused and masked refinements were required to attain the reconstructions of FluPol:NP-0:NP+1:NP+2:RNA unit, NP-0:RNA unit, and NP+1:NP+2:RNA unit (the positions for NP-0, NP+1, NP+2, NP+3, NP+4, NP-1, NP-2, and NP-3 are defined in Fig. 1) from State-In mini-vRNP at 2.97-Å, 4.75-Å, and 4.52-Å resolution, and the reconstructions of FluPol:NP-0:RNA unit and FluPol unit from State-Out mini-vRNP at 3.83-Å and 3.62-Å resolution (see Methods). The reconstruction of the empty ring did not generate an interpretable density due to severe preferential orientation bias during data collection. The atomic models were built according to the densities with local refinement. Refined models were fitted into their consensus positions in the unfocused reconstruction, allowing the overall architecture of the mini-vRNPs to be presented.

### Overall structures

In both State-In and State-Out mini-vRNPs, one FluPol and eight NPs were identified in the overall cryo-EM map (Fig. 1 and Supplementary Movies 1–4). Eight NPs pack side-by-side to form a ring-shaped arrangement, and the vRNA is bound in the inner side of the ring. The 5′ and 3′ termini of vRNA are bound together in FluPol in a manner similar to that previously reported[19]. One NP (NP-0) binds the fork of the double-stranded region of the vRNA that is distal from the 5′/3′ vRNA promoter, thus forming a FluPol:NP-0:RNA unit. For the State-In mini-vRNP, FluPol positions vertically above the NP-RNA ring and its PA CTD is enwrapped in the center hollow of the NP-RNA ring (Fig. 1a). In contrast, the FluPol:NP-0:RNA unit in State-Out, as a rigid body, has a ~90° rotation compared to its position in State-In, thus resulting in FluPol attaching at the outer rim of the NP-RNA ring (Fig. 1b, Supplementary Fig. 5a–c and Supplementary Movies 5 and 6). In both states, the relative orientation of FluPol, NP-0, and RNA is conserved (Supplementary Fig. 5d). Superimposition of FluPol:NP-0:RNA units in both states using FluPol as the reference shows that NP swings by ~15° to be alongside the double-stranded part of vRNA.

In both states, PA, PB1, and the first 250 residues of PB2 have unambiguous densities (Supplementary Fig. 6), whereas the densities corresponding to the C-terminal part of PB2 (residue 251 to the C-terminus) are only partially visible, indicating the C-terminal part of PB2 may be flexible relative to the other FluPol components. The structures of FluPol in State-In and State-Out share high similarities with an r.m.s.d of 0.723 Å for all the visible Cα atoms. The structures of NP-0 and NP+2, which have the best densities for the NPs in two representative positions, make structural movements in the D72-K90 and Q405-G460 regions (Supplementary Fig. 7). It is worth noting that the loop spanning residues D72-K90 (D72-K90 loop) of NP-0 folds into an α-helix upon binding to the fork of the double-stranded region of the vRNA, which is consistent with observation in other RNA-NP complex structures[20–22]. The D72-K90 loop in NP+2 undergoes structural movements compared to unbound RNA-NP structures, but is still maintained as a loop upon binding to linear vRNA, a phenomenon also observed in the previously determined NP and 3-mer RNA co-crystal structure (PDB ID: 7DXP)[23].

### Interaction of FluPol and NP

FluPol and NP mainly interact by NP-0 contacting FluPol, while the other NPs in the mini-vRNPs have fewer interactions with FluPol (Fig. 2a). Based on its best quality cryo-EM density, we used the model of the FluPol:NP-0:RNA unit in the State-In mini-vRNP to show the detailed interactions between FluPol and NP-0.

We observe that two loop regions in the head domain of NP-0 make interactions with PA and PB1 of FluPol (Fig. 2a). In region 1, a loop in NP-0 spanning the residues I201-N211, which links two helices in the head domain of NP-0, contacts residues of PA and PB1, as well as the nucleotides of the vRNA 5′ terminus, close to the entrance of vRNA 5′ terminus in FluPol (Fig. 2b). $_{NP-0}$R208 forms an elaborate network with other RNP components. Its side chain contacts the main chain atoms of

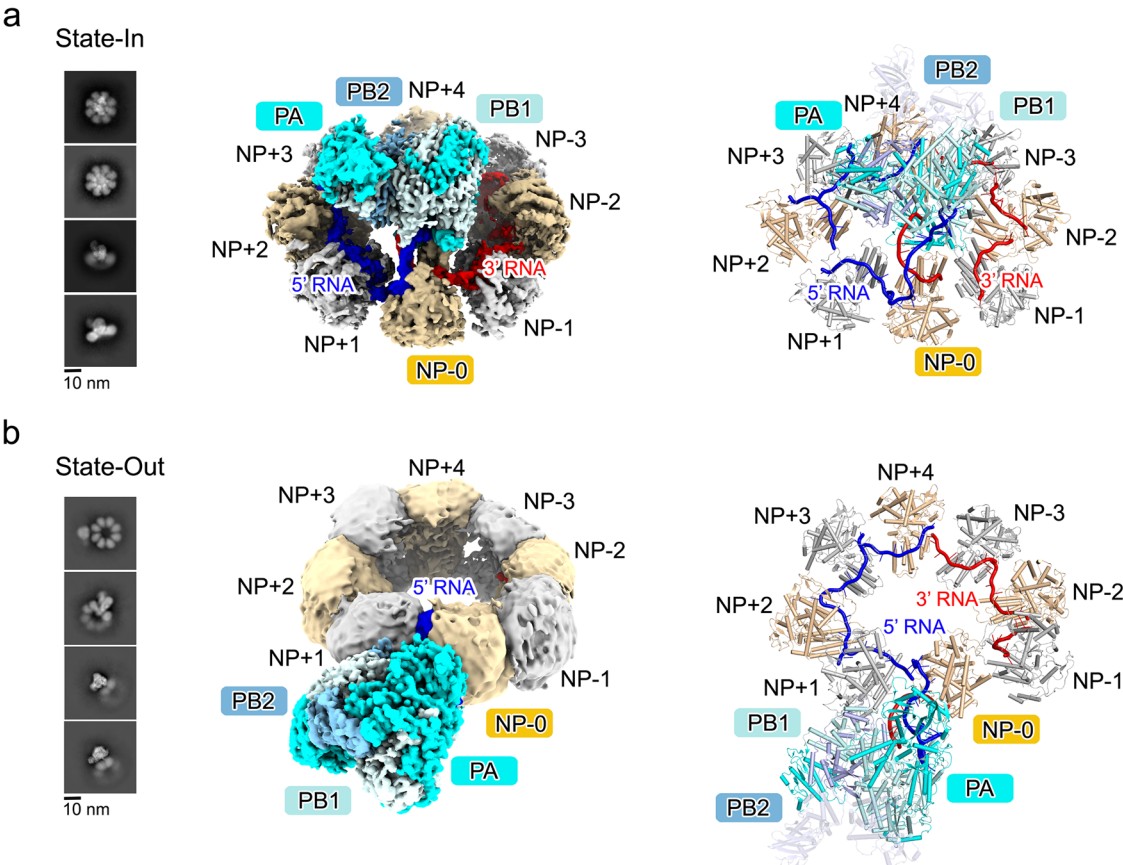

**Fig. 1 | Overall structures of mini-vRNPs.** The structures for the mini-vRNP in State-In and in State-Out forms are presented in (**a**, **b**), respectively. The representative 2D classifications are shown in the left panels. The cryo-EM densities for the mini-vRNPs are shown as isosurface rendering in the middle panels. The densities corresponding to PA, PB1, PB2, 3′ vRNA, or 5′ vRNA are colored cyan, pale cyan, light blue, red, and blue, respectively. NP-0 is colored in light yellow, while other NPs are displayed in white or light yellow. The models for the two states of the mini-vRNP are shown in the right panels as cartoon diagrams with the same color scheme as in the middle panels. The nucleotides bound in NP-0/+1/+2/+3 or in NP-1/-2/-3 are colored blue and red to indicate they are in the 5′ or the 3′ directions of vRNA, though they represent one single-stranded vRNA. Because the polypeptide of the PB2 C-terminal region (from residue 251 to the C-terminus) has only partially visible densities, the model of this part derived from a structure of influenza A virus polymerase (PDB code: 5M3H) is displayed as a semi-transparent cartoon.

$_{PA}$A375, $_{PA}$P376, $_{PA}$E377, and the side chain atom of $_{PA}$A375. In addition, its guanidyl group stabilizes the phosphate groups of rA+11 and rG+12 at the 5′ end of vRNA. In region 2, the loop region in NP-0 spanning residues E244-A251 has close contacts to the residues covering the region of K362-M374 in the CTD of PA (PA$_C$) (Fig. 2c). The side chain of $_{NP-0}$N247 hydrogen-bonds with the side chain atoms of $_{PA}$N373 and $_{PA}$K367, meanwhile the main chain atom of $_{NP-0}$N247 also has a hydrogen-bond mediated interaction with the side chain of the $_{PA}$K367. Additionally, $_{NP-0}$S245 makes minor contacts with $_{PA}$S364 to stabilize the interaction of FluPol and NP. These key residues involved in the interaction between NP and PA in mini-vRNP are highly conserved in representative strains of influenza A virus, except for bat H17N10 and H18N11 strains (Supplementary Fig. 8).

To examine the impact of these interacting residues on replication and transcription, we performed mini-vRNP reconstitution assays in 293T cells with RNA harvested 12 h post-transfection (Fig. 2d−f and Supplementary Fig. 9). The results show that NP R208A mutation is strongly inhibitory of mini-vRNP function, while other mutations on PA or NP (including S364A, K367A, N373A, E377A of PA and S245A, N247A of NP) have no inhibitory effect on vRNP activity. Previous studies have shown that the residues in region 1 play essential roles in influenza virus replication and/or transcription, including R208[24,25], which is consistent with the structural and biochemical observations here that this region plays the most essential role in interacting with FluPol to form RNP. The arginine at position 208 (R208) in NP is highly

conserved in representative influenza A virus strains, except in bat H17N10 and H18N11 strains, where it is replaced by lysine (Supplementary Fig. 8). The lack of inhibitory effect of other mutations suggests the FluPol-NP interactions in vRNPs are likely to be highly dynamic and the observed interface might represent one of multiple possible polymerase-NP arrangements that is stable enough to be captured by cryo-EM.

## Contact of RNA with FluPol

The interactions of the vRNA 5′ and 3′ termini with FluPol in the State-In and State-Out structures are generally similar to those observed in previous structures[19]. The first ten nucleotides of the 5′ end of the vRNA (rA+1 to rA+10) form a compact stem-loop fold that is inserted into a deep pocket at the interface of the PB1 fingers and the PA CTD (Fig. 3a, b). Nine nucleotides of the 3′ end of vRNA (rU+239 to rC+247) bind nearby in a region that includes residues from all three polymerase subunits. The nucleotides of the 5′ end of vRNA (rA+10 to rG+16) and the nucleotides of the 3′ end (rC+234 to rU+239) form a double-stranded region at the distal part of the vRNA promoter, which projects away from the main body of FluPol and extends to NP-0.

## Interaction of RNA with NPs

The interaction of NP and vRNA has two distinct forms, either in FluPol:NP-0:RNA unit, or in NP+1:NP+2:RNA unit representing the vRNA in complex with NPs in the regular positions (Figs. 3c−e and 4 and

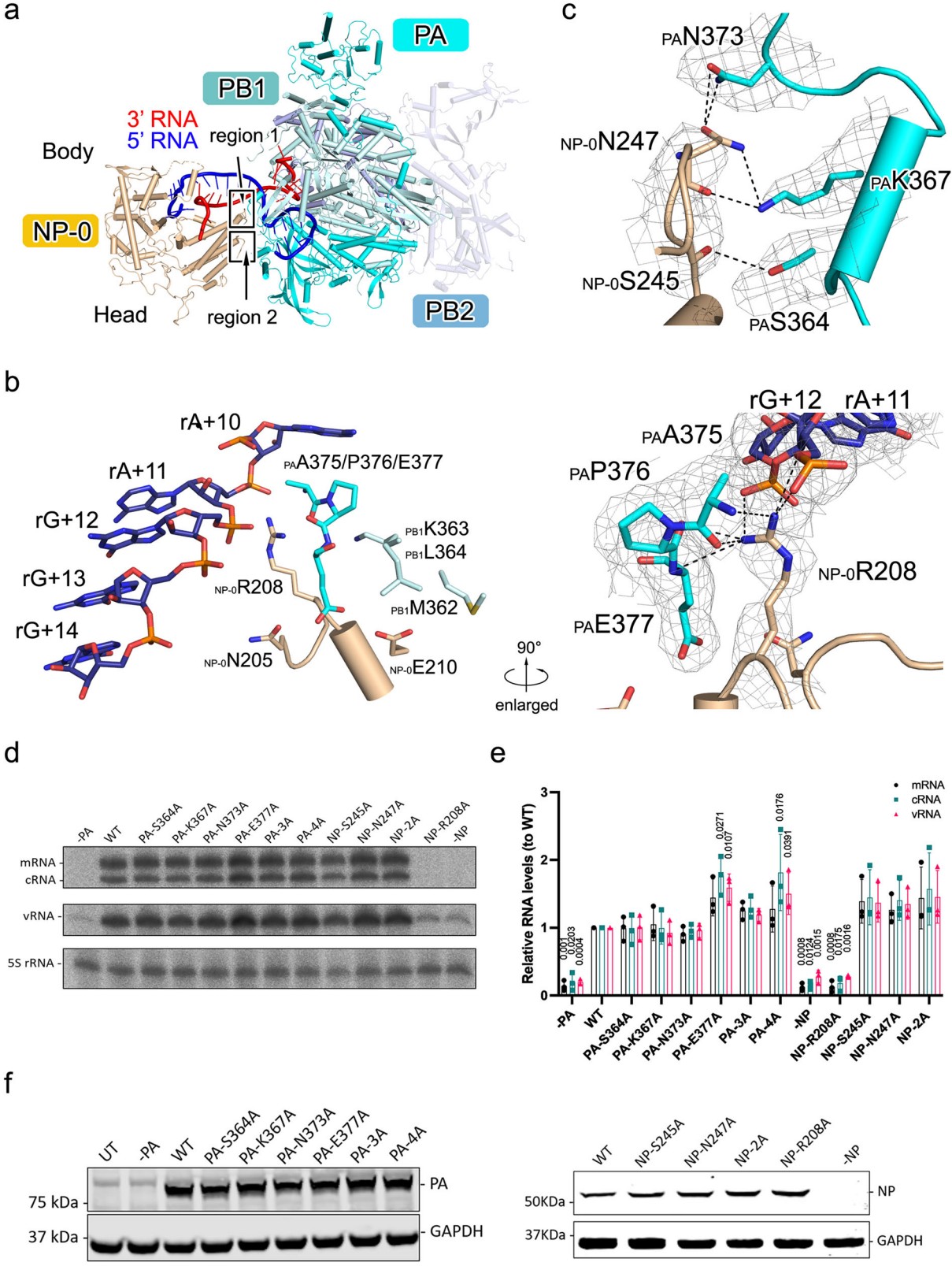

Supplementary Fig. 10). Compared to NPs in the regular position (e.g., NP+2), two regions of the polypeptide of NP-0 (D72-K90 and I201-N211) exhibit a clear conformational shift upon binding with RNA and FluPol, in particular the D72-K90 loop that folded into an α-helix (Supplementary Fig. 7a).

In the FluPol:NP-0:RNA unit, the nucleotides at the 5′ and 3′ ends of vRNA form a distal double-stranded section of the vRNA promoter,

which extrudes out from FluPol and points to NP-0. The loop spanning residues D72 to K90 in NP-0, which includes an α-helix region (D72-E81), inserts into the fork of the double-stranded part of the paired vRNA, thus separating them into single strands (Figs. 3c and 4a–c). The single-stranded RNA in both 5′ and 3′ directions is further bound into a deep positively charged groove clamped by the head and the body domain of NP-0, forming an RNA-binding groove to protect the RNA

**Fig. 2 | Interactions of FluPol and NP. a** A side view of the structure of FluPol:NP-0:RNA is shown as a cartoon diagram with the same color scheme as used in Fig. 1. The body and head domains of NP-0 are indicated. The polypeptide of the C-terminal part of PB2 is displayed as a semi-transparent cartoon. **b** A close-up view of region 1 on the interface of FluPol and NP-0. Key residues and the nucleotides of 5′ vRNA are shown as colored sticks. The interactions surrounding $_{NP-0}R208$ are enlarged and exhibited in the right panel. Key residues and the nucleotides are covered by cryo-EM densities as a gray mesh. Inter-molecular interactions are denoted as dashed lines. **c** A close-up view of region 2 on the interface of FluPol and NP-0. Key residues are shown as colored cartoons, and the side chains are shown as colored sticks. Key residues are covered by cryo-EM densities as gray mesh. Intermolecular interactions are denoted as dashed lines. **d** Effect of PA and NP mutations on influenza A/Victoria/3/75 (H3N2) mini-vRNP activity. A neuraminidase (NA) vRNA template was used. Total RNA was isolated at 12 h post-transfection and analyzed by primer extension. WT, wild type; PA-3A, PA-S364A/K367A/N373A; PA-4A, PA-S364A/K367A/N373A/E377A; NP-2A, NP-S245A/N247A. **e** Quantification of viral RNAs by primer extension. Error bars represent the standard deviation from three independent experiments. Data are presented as mean values ± SD. **f** Western blot analysis of the expression of mutant proteins in mini-vRNP reconstitution assays. **d, f** Source data are provided as a Source data file.

(Supplementary Fig. 10a). The interacting residues of 5′ RNA in NP-0 include L61, R65, G86, K87, N144, T147, Y148, Q149, T151, R152, R208, R355, G356, R361, G362, Q364, A366 and S367. The residues of NP-0 that contact with the 3′ RNA include R74, Y78, H82, R156, T157, R174, R195, R195, M196, R199, F206 and K214.

In the NP+1:NP+2:RNA unit that represents the RNA encapsidated by NP in the regular position of RNP, the RNA-binding grooves of each adjacent NP form a continuous path and play a dominant role in stabilizing the vRNA (Figs. 3d, e and 4d–f and Supplementary Fig. 10b). Unlike NP-0 which binds the fork of the double-stranded part of the vRNA, each NP in the regular position accommodates ten nucleotides in a linear sequence and the RNA bound in RNA-binding grooves locates on the inner side of the NP ring. The D72-K90 loop in NP+2 does not fold into an α-helix; it undergoes a conformational change to adopt a more open loop conformation positioned above the linear RNA. This structural shift enables broader surface coverage and stabilization of the vRNA, thereby protecting it from degradation. The residues that are close to the linear RNA in the regular-positioned NPs include R65, E73, N76, Y78, E80, S84, K87, Y148, R150, S176, R199, R221, K214, R361, and Q364. RNA sequestering by influenza virus NPs is likely a common mechanism used by negative-strand RNA viruses to protect their genomes[26]. Notably, the RNA-binding grooves of NPs in the regular positions face the inner side of the NP-RNA ring (Fig. 1), thus burying a previously described bipartite NLS loop of NP[27], suggesting that this bipartite NLS is not likely to play a role in the nuclear transport of RNP, in agreement with Cros et al.[28].

Among the residues in the RNA-binding groove of NP, previous studies have shown that mutations of G93, Y148, R152, R156, R174, R195, R199, R213, and R361 to alanine completely eliminate virus proliferation, while mutations of R65, R74, R/K214, and R221 attenuate virus proliferation[24]. This is consistent with their roles in RNA binding and forming the mini-vRNP. A previous crystallographic study has shown that monomeric NP binds RNA in a positively charged pocket mainly in the body domain[23], which is different compared with the RNA-binding groove observed in the structures of the mini-vRNPs (Supplementary Fig. 10c). However, the recently published cryo-EM structures reveal that the RNA is located in the RNA-binding groove between the head and body domains of NP[20–22], which is consistent with our observations (Supplementary Fig. 10e–h). Meanwhile, the RNA is also observed located at the interface of neighboring NP in these cryo-EM structures, a detail that is invisible in our structures, suggesting that the conformation of NP and RNA within the mini-vRNP is flexible. Contrastingly, G490-D497 at the C-terminus of NP partially occupy the RNA-binding groove identified in the mini-vRNP (Supplementary Fig. 10d). Since the C-terminal residues of NP are not visible in the mini-vRNP, as well as in other influenza virus NP-RNA complex of reconstituted RNP-like helical or RNP complex[20–22], we propose that this region may undergo a significant conformational shift upon the formation of the ring-shaped NP oligomer, thus releasing the RNA-binding groove to adopt viral RNA. A similar conformational shift of terminal residues of nucleoprotein upon oligomerization for RNP formation has also been observed in other negative-strand RNA viruses[29,30]. It is also noteworthy that although NPs stack side-by-side to bind the continuous vRNA, direct interaction among core domains of NPs is not observed in our models (Supplementary Fig. 11). The minimal distances of NP-0:NP+1 and NP+1:NP+2 range from 10–15 Å, suggesting the inter-NP orientations may be flexible, possibly necessary for forming native RNPs.

## Discussion

The atomic structures of influenza virus mini-vRNPs reported here provide insights into the molecular details for the coupling of polymerase, nucleoprotein, and RNA in mini-vRNP, as well as provide a model system to further understand the architecture of influenza virus mini-vRNP. Although the interactions of polymerase, nucleoprotein, and RNA are elucidated at the atomic level, in this work, additional information underlying the filamentous part of RNP with helical features warrants further investigation. Through the mini-vRNP structures, we discovered that NP-0 serves as the primary FluPol interaction site within mini-vRNPs, while other NPs show minimal FluPol contacts (Fig. 2). Notably, the residue R208 of NP-0 forms an extensive interaction network with other mini-vRNP components, which is crucial for NP-FluPol association during RNP assembly. And the arginine 208 in NP is highly conserved in representative influenza A virus strains, except in bat H17N10 and H18N11 strains, where it is replaced by lysine (Supplementary Fig. 8). Additional, the distal 5′ and 3′ vRNA duplex are stabilized and separated by NP-0, which is not consistent with the previous vRNP model that proposed the 5′ and 3′ distal ends would attach one NP molecular, respectively[18,22]. Regarding this, we think that NP-0 is essential to the mini-vRNP, which stabilizes the circularization of the NP-RNA ring scaffold in both the inert state and the active state. Additionally, it can better shield the terminal vRNA from degradation during the viral life cycle. However, it is noteworthy that in the reconstituted RNP structures, the element of NP-0 bifurcated RNA was not observed, possibly due to the resolution limitations[22]. Therefore, we cannot exclude the possibility that the native RNPs may not adopt the NP-0 element. This implies that the near-atomic resolution structures of native or reconstituted vRNPs with full architecture are urgently demanded.

Recently, several cryo-EM structures of recombinant helical NP-RNA complex, reconstituted RNPs, and native RNPs were reported[20–22]. Consistent with our observations, the RNA-binding groove in NPs is located between the head and body domains. The RNA direction (5′ to 3′), RNA path, and the RNA-binding sites in NPs also roughly match with those observed in our structure of mini-vRNP. The confirmed RNA-binding groove in NP would be a potential target for anti-influenza therapeutic development. In these structures, the RNA is also observed located at the interface between two neighboring NPs, while a large open pocket could connect the RNA fragments from two NP subunits. This feature, not resolved in our structures, suggests conformational flexibility in both NP and RNA within the mini-vRNP assembly. Additionally, an RNA-binding-induced loop-to-α-helix turn (region D72-K90) in influenza virus NPs was observed in these structures. In our structures, the D72-K90 loop of NP-0 folds into an α-helix when bound to the RNA fork, but in NP+2 the D72-K90 loop adopts a more open loop upon binding to the linear RNA. This difference suggests the flexible D72-K90 loop is crucial for RNA binding, and flexible NP orientations that may be necessary for native RNP assembly.

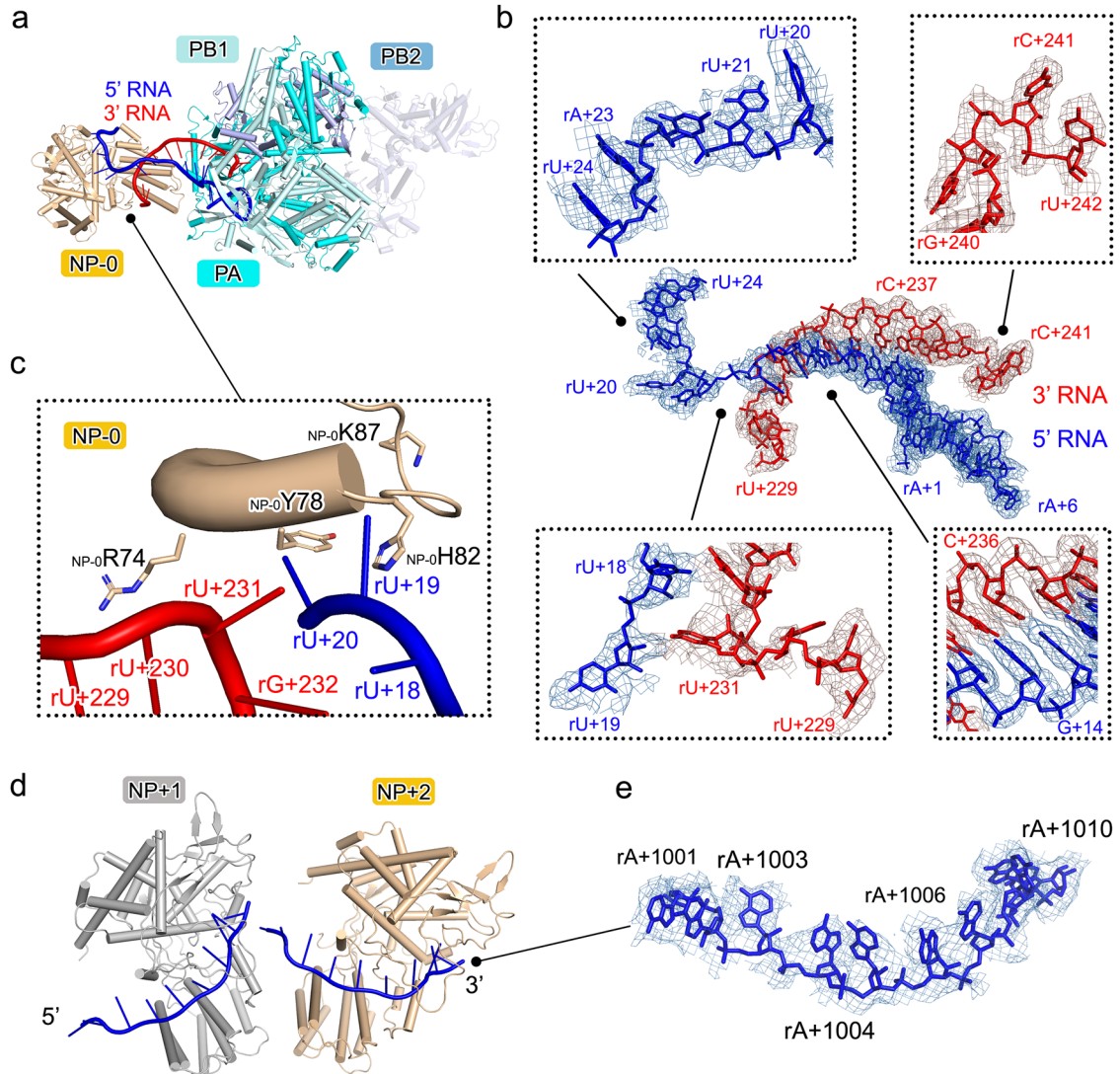

**Fig. 3 | Interactions of RNA with FluPol and NPs. a** Top view of the structure of FluPol:NP-0:RNA of the mini-vRNP in State-In is shown as a cartoon diagram with the same color scheme as used in Fig. 1. **b** The nucleotides bound in FluPol and NP-0 are shown as colored sticks and are covered by cryo-EM densities as colored mesh. An overall view is shown in the center of (**b**), and four close-up enlarged views are displayed in the frames. Several nucleotides are labeled to identify their positions. **c** An enlarged view of the D72-K90 loop region of NP-0 that inserts into the fork of the paired 5′/3′ vRNA. The residues R74, Y78, H82, and K87 of NP-0 are shown as colored sticks. The nucleotides at the fork of the double-stranded part of vRNA are labeled. **d** The structure of NP+1:NP+2:RNA unit. The polypeptides of NP+1 and NP +2 are presented as white and light yellow cartoons. The bound RNAs are shown as blue cartoons. The 3′ and 5′ directions of the bound RNA are indicated. **e** The nucleotides that bind to NP+2 are displayed as blue sticks and are covered by cryo-EM densities as blue mesh. Several nucleotides are labeled to identify their positions.

Although the reconstituted mini-vRNP assembly used a 248 nt vRNA, the resolved structures revealed significantly less bound vRNA. Specifically, we observed approximately 38 nt of vRNA in the State-In FluPol:NP-0:RNA unit and about 10 nt in the State-In NP+2 unit. The 5′ and 3′ terminal region of the vRNA, which participates in interactions with FluPol and forms a double-strand duplex, comprises approximately 30 nt, which suggests each NP likely binds approximately 24–28 nt of vRNA, and the number of nucleobases that binds to NP is not fixed. In the NP+2 unit, the density allowed us to model 10 nt within the groove; we propose that the remaining nucleotides are attached to the surface between adjacent NPs. Recent reports suggest that large open pockets exist at NP-NP interfaces: Peng et al. speculate a pocket atop the linker region connecting RNA fragments from adjacent NPs[22], while Chenavier et al. propose that the NP's lateral surface accommodates a variable number of nucleobases[21]. This structural flexibility suggests viral RNA in these regions may form protruding stem-loop structures, which

could mediate different RNP segment interactions essential for precise viral genome packaging[31,32].

Remarkably, the interface for host acidic nuclear phosphoprotein 32 (ANP32) mediated FluPol dimerization (located in the 432–438 loop of the PA, the N2, mid-link subdomain and 627 domain of PB2 for replicase) that is required for genome replication[10–12] and the interface for FluPol association with Pol II (located in the CTD of the PA subunit) that is required for transcription initiation[13,14] are obstructed by the NP-RNA ring in the State-In structure, while they are fully exposed in the State-Out structure (Supplementary Fig. 12). This observation suggests that the State-In structure may represent an inert state of mini-vRNP, and a conformational change of FluPol:NP-0:RNA unit would precede viral replication or transcription. Moreover, in the mini-vRNPs that we purified, the State-In conformation accounts for a larger proportion than the State-Out conformation, which is consistent with the observation that in other vRNP research, inert vRNPs account for a higher proportion than active vRNPs[18].

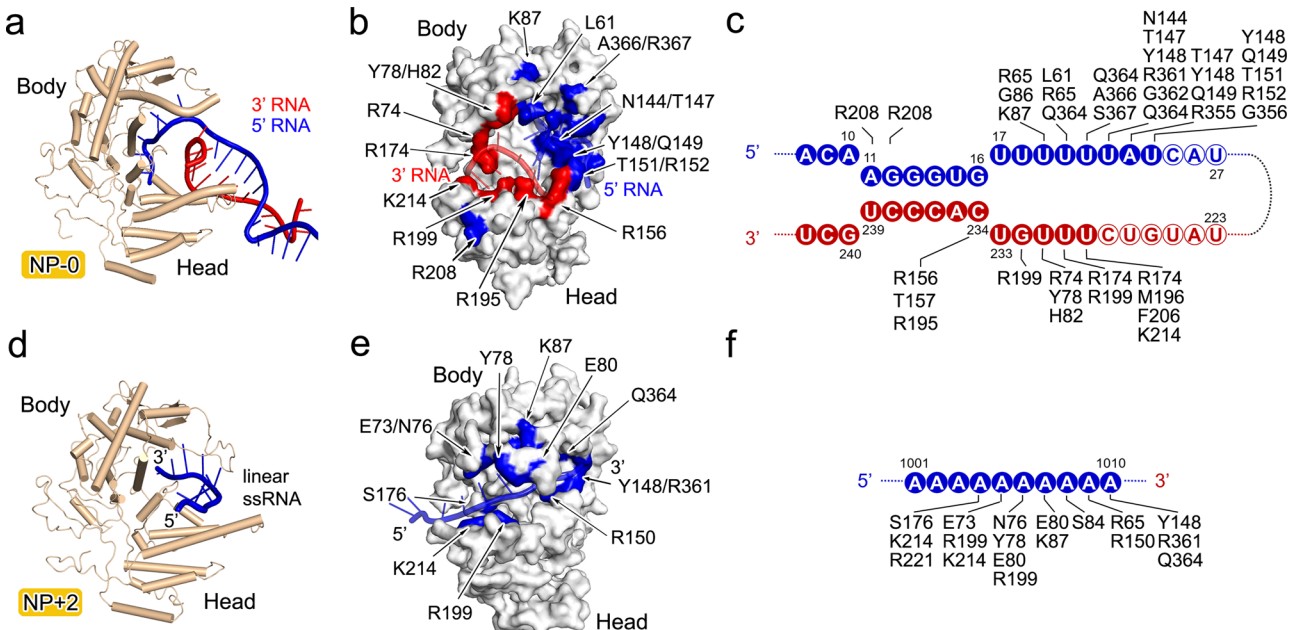

**Fig. 4 | Interactions of RNA with NP-0 and NP+2.** The interaction of RNAs with NP-0 and NP+2 is analyzed in (**a**–**f**), respectively. **a**, **d** The structures of NP-0:RNA and NP+2:RNA are shown as cartoon diagrams with the same color scheme as in Fig. 1. **b**, **e** The molecules of NP-0 and NP+2 are depicted as molecular surface in white. The key residues of NP-0 that interact with 3′ RNA or 5′ RNA are colored as red and blue on the surface (**b**). The key residues of NP+2 that interact with RNA are colored as blue on the surface (**e**). The RNA molecules are shown as semi-transparent cartoons. Schematic of the protein-RNA interactions for NP-0:RNA (**c**) and NP+2:RNA (**f**). Solid and hollow circles indicate nucleotides that were visible or invisible, respectively. The amino acid residues interacting with each nucleotide (within 4.0 Å) are identified and listed next to their interacting nucleotides.

## Methods

### Plasmids and cell transfection

To construct plasmids expressing mini-vRNP components in mammalian cells, the genes encoding PA, PB1, PB2, and NP of influenza A/Victoria/3/75 were cloned into a pCAG vector, respectively, to generate the plasmids pCAG-PA, pCAG-PB1, pCAG-PB2, and pCAG-NP, among which pCAG-PB2 possessed a C-terminal 8xHis-tag for the affinity purification of mini-vRNP. A previously reported 248 nt ΔNS sequence derived from the NS gene of the virus[33], which was promoted by a human Pol I promoter and terminated by a murine Pol I terminator, was also cloned into the pCAG vector to encode a vRNA, generating the plasmid pCAG-vRNA.

For cell transfection, the HEK293F cells were first cultured up to $2 \times 10^6$ cells/mL in SMM 293 TII medium (SinoBiological, China). A mixture of the plasmids containing (for 500 mL cell cultures) pCAG-PA (90 μg), pCAG-PB1 (90 μg), pCAG-PB2 (170 μg), pCAG-NP (90 μg), and pCAG-vRNA (60 μg) was diluted to 7.5 mL in SMM 293 TII medium, and in another tube, 2 mL polyethylenimine (PEI) at 1 mg/mL was diluted to 7.5 mL in SMM 293 TII medium. The solutions from the two tubes were then mixed and kept at room temperature for 20 min and supplemented to 500 mL cell cultures. The cells were cultured for 48 h after transfection.

### The purification of the mini-vRNP

Cultures of HEK293F cells transfected as above were harvested by centrifugation, and were resuspended in lysis buffer (10 mM Tris-HCl, 1 mM EDTA, 7.5 mM $(NH_4)_2SO_4$, 1 mM TCEP, 0.025% NP40, pH 8.0) and lysed at 4 °C. The insoluble material was removed by centrifugation at 13,000 rpm for 40 min at 4 °C. For RNP purification, the supernatant was first purified by Ni-NTA affinity chromatography. The resin was washed with 100 volumes of buffer containing 50 mM Tris-HCl, 100 mM NaCl, 5 mM MgCl₂, 1 mM TCEP, 0.1% NP40, 20 mM imidazole at pH 8.0, followed by another five column volumes of the same buffer containing an additional 40 mM imidazole and another five column volumes of the same buffer containing an additional 60 mM imidazole. After that, the mini-vRNPs were eluted with three column volumes of 50 mM Tris-HCl, 100 mM NaCl, 5 mM MgCl₂, 1 mM TECP, 0.1% NP40, 80 mM imidazole, EDTA-free protease inhibitors cocktail at pH 8.0, another five column volumes of the same buffer containing 150 mM imidazole, and another three column volumes of the same buffer containing 500 mM imidazole. Finally, the extract was centrifuged on a 15–35% glycerol gradient in the buffer containing 50 mM Tris-HCl, 100 mM NaCl, 5 mM MgCl₂, 1 mM TCEP, 0.1% NP40, EDTA-free protease inhibitors cocktail at pH 8.0 for 17 h with 35,000 rpm at 4 °C in a SW41 rotor. The purified RNPs were concentrated to 0.4 mg/mL and then used for further analyses.

### Fluorescence analysis

The biological activity of the reconstituted RNPs in the expression cells was verified through GFP expression and fluorescence observation. To construct plasmid pCAG-eGFP, a reverse eGFP gene was inserted into the ΔNS sequence of plasmid pCAG-vRNA, with 5′ and 3′ conserved termini of the NS sequence retained. Cultures of COS-1 cells in Dulbecco modified Eagle medium (DMEM) with 10% fetal bovine serum (FBS) were transfected with the mixtures of the plasmids containing (for 100-mm dishes) pCAG-PA (1.5 μg), pCAG-PB1 (1.5 μg), pCAG-PB2 (3 μg), pCAG-NP (1.5 μg), and pCAG-eGFP (1 μg). The plasmid mixtures were combined with 17 μL of Lipofectamine 2000 Reagent (Invitrogen, USA), and diluted by adding 200 μL of Opti-MEM (Gibco, USA) separately, and were then mixed and kept at room temperature for 15 min before adding to the culture plates containing 5 mL of Opti-MEM. After 5 h of incubation at 37 °C, the medium was replaced by 10 mL of DMEM with 10% FBS and further incubated for 0–72 h. During this time, the culture plates were subjected to fluorescence observation.

### Real-time PCR

Total RNA of purified and concentrated RNP samples was extracted by the FastPure Cell/Tissue Total RNA Isolation Kit-BOX2 (Vazyme,

China). Reverse transcription was then performed by HiScript II 1st Strand cDNA Synthesis Kit (Vazyme, China) using tagged primers to distinguish vRNA and cRNA with high specificity (Fig. S2A)[34]. To plot standard curves, synthetic vRNA or cRNA was first generated by RiboMAX Large Scale RNA Production Systems-T7 (Promega, USA), followed by RQ1 DNase I (Promega) digestions, in which a linear DNA with T7 promoter generated by PCR was used as the template, and then purified by HiPure RNA Pure Micro Kit (Magen, China). Concentrations of purified vRNA or cRNA transcripts were determined by spectrophotometry and diluted to a series of ten-fold copies, respectively, to form RNA standards, which were reverse transcribed using the same method as RNP samples afterwards. Real-time PCR (qPCR) was performed with ChamQ Universal SYBR qPCR Master Mix (Vazyme, China) on a QuantStudio 1 system (Thermo Fisher). The primers used for qPCR are listed as below: 5′-GGCCGTCATGGTGGC-GAAT-3′ and 5′-CTAGACCGAGAGTGCTGCCTCTTCC-3′ for vRNA, 5′-GCTAGCTTCAGCTAGGCATC-3′ and 5′-AAAAGTTCGAAGAGATAAGATG GC-3′ for cRNA.

### Cryo-EM sample preparation and data collection

For cryo-EM sample preparation, 5 μl aliquots of purified mini-vRNPs at 0.4 mg/mL were applied to Quantifoil R1.2/1.3 or R2/1 using ultrathin carbon grids (Quantifoil, Micro Tools GmbH, Germany) previously glow discharged in low air pressure, blotted for 3.5 s with a blot force of 1 at 8 °C and 100% humidity, and plunge-frozen in liquid ethane using a Vitrobot (Thermo Fisher Scientific, USA)[35]. For the State-In mini-vRNP sample, the cryo-EM data were collected with a 300 kV Titan Krios electron microscope (Thermo Fisher Scientific, USA) and a Falcon4 direct electron detector (Thermo Fisher Scientific, USA). A series of micrographs were collected as movies and recorded with a −2.0 to −1 μm defocus at a calibrated magnification of 130,000×, resulting in a pixel size of 0.96 Å per pixel. For State-Out mini-vRNP, the cryo-EM data were collected with a 300 kV Titan Krios electron microscope (Thermo Fisher Scientific, USA) and a K3 direct electron detector (Gatan, USA). A series of micrographs were collected as movies and recorded with a −2.0 to −1 μm defocus at a calibrated magnification of 22,500×, resulting in a pixel size of 1.06 Å per pixel. The exposure time was set to 2 s with a total accumulated dose of 60 electrons per Å$^2$. All images were automatically recorded using SerialEM[36]. Statistics for data collection and refinement are in Supplementary Table 1.

### Cryo-EM data processing

For processing the cryo-EM data of the mini-vRNP in State-In, each movie was motion corrected and dose weighted by MotionCor2[37]. The contrast transfer function (CTF) was estimated by patch CTF in CryoSPARC v3.2.0[38]. A total of 13,796 micrographs were selected for further data processing. All subsequent analyses were performed in cryoSPARC except for those specifically mentioned. We first picked 500 micrographs through blob picker, and the extracted particles were subjected to one round of 2D classification. All classes with clear features of State-In mini-vRNP were selected as templates for further particle picking. Subsequently, we used template picker with different particle diameters (200 Å and 300 Å) to automatically pick particles and extracted particles with the box size of 360 pixels. After multiple rounds of 2D classification, a total of 1,099,174 particles with the features of State-In mini-vRNP were selected and combined, and duplicates were removed. Due to the highly heterogeneous dataset, an approach similar to "Build and Retrieve" was employed for particle sorting. Briefly, we optimized the initial 3D maps using ab initio methods, and the particles with different features were retrieved from the complete dataset based on these maps. To obtain the initial map, three repetitions of ab inito were executed ($K = 6$, where $K$ is the number of classes), and the particles from the best initial model were selected to execute another round of ab inito ($K = 3$) building. These volumes were further optimized using heterogeneous refinement. One

of these maps displayed clear secondary structure information. To increase the number of particles, we use these maps to retrieve particles from the complete dataset using heterogeneous refinement. This resulted in 265,190 clean particles for NU-refinement, ultimately yielding a State-In mini-vRNP structure with an overall resolution of 2.89 Å. The local resolution of the final map was estimated in cryoSPARC.

To investigate the detailed interaction within FluPol:NP:RNA, the particles were subtracted and performed local refinement focused on the FluPol:NP-0: NP+1:NP+2:RNA region. Subsequently, a round of non-alignment 3D classification was performed, and the best aligned class was selected for local refinement. This resulted in a better density for NP-0. However, due to the flexibility between NP molecules in the NP-RNA ring, the densities for NP+1, NP+2, and NP-1 remained unresolved. To further reduce heterogeneity, we applied one round of non-alignment 3D classification ($T = 16$, $K = 5$) in RELION3.1.2[39], masking NP +1 and NP+2, and selected the best class for subsequent local refinement in cryoSPARC. The resolutions for the final local maps were as follows: FluPol:NP-0:NP+1:NP+2:NP-1:RNA, 2.97 Å; NP-0:RNA, 4.75 Å; NP+1:NP+2:RNA, 4.52 Å.

For processing the cryo-EM data of the mini-vRNP in State-Out, a total of 25,550 micrographs were selected after motion correction and CTF estimation. We first picked 1000 micrographs through blob picker, and the picked particles were subjected to one round of 2D classification. All classes with the features of State-Out mini-vRNP were selected as the templates for further particle picking. Subsequently, the particles were automatically picked by template picker with different diameters (360 Å and 400 Å) from all micrographs and extracted with a box size of 440 pixels. Multiple rounds of 2D classification were applied for each of the separately extracted particles, and a total of 2,267,710 particles with the features of State-Out mini-vRNP were selected and combined, and the duplicates were removed. Following the same particle sorting strategy as mentioned above, 112,185 clean particles were selected for NU-refinement, yielding the overall map of State-Out mini-vRNP with a nominal resolution of 5.54 Å.

To further improve the local densities in the State-Out mini-vRNP structure, the particles were subtracted and local refinement focused on FluPol:NP-0:RNA was performed, yielding an improved local density with the resolution of 3.83 Å. We further masked the FluPol region to give a final resolution of 3.62 Å.

Automatic local sharpening in DeepEMhancer[40] performed on all maps significantly improved the interpretability of local density. Locally refined maps were combined into the global map using the "Combine Focused Maps" strategy in PHENIX[41]. The complete workflow is shown in Supplementary Figs. 3 and 4.

### Model building and refinement

To build the structure of the RNP complex, the coordinates of the influenza A virus polymerase (PDB code: 5M3H)[13] and influenza A virus NP (PDB code: 3ZDP and 7NT8)[42,43] were individually placed and fitted by rigid-body refinement into the cryo-EM maps using UCSF ChimeraX[44]. The model was manually rebuilt with the guidance of the cryo-EM map using Coot[45], and in combination with real-space refinement using Phenix[41]. The data validation statistics are shown in Supplementary Table 1.

### Transfection for vRNP reconstitution and primer extension analysis of viral RNA

293T cells in 24-well plates were transfected with pcDNA3.1 plasmids containing the coding regions for the A/WSN/33 (H1N1) (WSN) polymerase genes PB2, PB1 and PA at 200 ng each[46]. The WSN NP-encoding plasmid was transfected at 400 ng. For A/Victoria/3/75, the pCAG-derived plasmids described above were transfected at identical amounts to WSN. A pPolI plasmid encoding the WSN-NA influenza vRNA was transfected at 100 ng[47]. Lipofectamine 2000 (Thermo Fisher

Scientific Inc.) was used for transfection according to the manufacturer's protocols. Transfected plates were incubated at 37 °C. After 12 h post-transfection, 500 µl of TRI Reagent® (Sigma) was added, and total RNA was extracted following the manufacturer's protocols. First-strand RNA synthesis via SuperScript III (Thermo Fisher Scientific Inc.) was conducted using NA-specific [$\gamma-^{32}$P] ATP (PerkinElmer) radiolabeled primers[46,48]. A primer specific to 5S rRNA was used as an internal control. Products were resolved on a 6% or 12% denaturing PAGE gel with 7 M urea. The resulting gels were imaged by phosphor imaging through a FLA-5000 scanner (Fuji), and targets were quantified using GelAnalyzer (V 19.1). Quantified results were normalized to the 5S rRNA signal and then plotted relative to the WT signal using Prism (Version 10.1.0). Experiments were done in triplicates, and statistical analysis was performed by one-way ANOVA.

### Expression of mutant PA and NP in vRNP reconstitution assays

Cells were lysed using RIPA lysis buffer (50 mM Tris, pH 7.5, 150 mM NaCl, 0.1% sodium dodecyl sulfate (SDS), 0.5% sodium deoxycholate, 1% Triton X-100, 5 mM DTT, and cOmplete™, Mini, EDTA-free Protease Inhibitor Cocktail). Samples were separated by SDS-PAGE and blotted onto 0.45 µM Nitrocellulose membranes (Amersham). Primary antibodies for anti-NP (GTX125989) from GeneTex and a previously described anti-PA antibody[49] were used. The primary anti-GAPDH antibody was supplied by Cell Signalling Technology (14C10). Secondary, conjugated IRDye 800CW goat anti-rabbit IgG (Li-COR 926-32211) and IRDye 680LT anti-mouse IgG (Li-COR 926-32220) antibodies were used before detection with the Li-COR Odyssey DLx platform. Images were processed with the affiliated software (LI-COR, Inc., 2022).

### Reporting summary

Further information on research design is available in the Nature Portfolio Reporting Summary linked to this article.

## Data availability

The cryo-EM density maps and the structures were deposited into the Electron Microscopy Data Bank (EMDB) and Protein Data Bank (PDB) with the accession numbers EMD-64766 and 9V44 for FluPol_NP-0_RNA of State-In, EMD-64770 and 9V48 for NP-0_RNA of State-In, EMD-64769 and 9V47 for NP+1_NP+2_RNA of State-In, EMD-64768 and 9V46 for FluPol of State-Out. The previously published influenza structures used in this study are available in the Protein Data Bank under accession codes 7DXP, 5M3H, 3ZDP, and 7NT8. Source data are provided with this paper.

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

## Acknowledgements

This work was supported by the Major Project of Guangzhou National Laboratory (GZNL2023A01002, Z.L.), the National Natural Science Foundation of China (32188101, Z.L. and 32300592, Y.Y.), the Research Fund-Vanke School of Public Health, Tsinghua University (2023JC002, Z.L.), Young Elite Scientists Sponsorship Program by CAST (2023QNRC001, Y.Y.), Beijing Natural Science Foundation (L242107, Y.Y.), and UK Medical Research Council (MR/X008312/1, E.F.). The authors thank the Bio-Electron Microscopy Facility of Shanghai Tech University and the Tsinghua University Branch of the China National Center for Protein Sciences (Beijing) for the cryo-EM facility.

## Author contributions

Z.L., Z.R., and E.F. conceived the project and designed the experiments. H.K., Y.Y., M.L., L.Z., Y. Lin, W.S., and Z.X. performed the mini-vRNP purification and biochemical study. H.K., L.Y., Y.H., and J.G. for biochemical assays. L.W. and K.Y.C. for minigenome assays. Y. Liu, L.Z., Y. Lin, Y.G. collected cryo-EM data. H.K., Y.Y., Y. Liu, L.Y., Z.R., Z.L., L.W., and K.Y.C. analyzed the data. Y. Guo and X.H. made in-depth discussion. H.K., Y.L., L.W.G., Z.R., and Z.L. wrote the manuscript. All authors discussed the experiments, read and approved the manuscript.

## Competing interests

The authors declare no competing interests.
