## [Transparent Peer Review file · Nature Communications]

Coupling of polymerase-nucleoprotein-RNA in an influenza virus mini ribonucleoprotein complex

Corresponding Author: Professor Zhiyong Lou

Version 0:

Reviewer comments:

Reviewer #1

(Remarks to the Author)

The authors present a robust and impactful study on the structure and function of the mini ribonucleoprotein (RNP) complex of the influenza virus. Recombinant molecules of NP, PB1, PB2, PA, and mini vRNA were expressed and used to purify mini-RNPs for structural and biochemical analyses. Techniques such as cryo-electron microscopy, RT-PCR, and mutagenesis RNA system assays were employed to elucidate the structure and function of mini-RNPs.

The study reveals that the mini-RNP complex exists in two major conformational states: State-In and State-Out. This classification is based on the relative position of the viral polymerase in relation to the NP ring. A series of interactions were identified, including NP–NP, NP–viral polymerase, NP–RNA, and viral polymerase–RNA, along with the presentation of viral protein interfaces that interact with cellular proteins. Based on the structural data, the authors propose a model in which molecular interactions and the positioning of the viral polymerase modulate mini-RNP RNA synthesis. This work not only advances our understanding of the structure–function relationship in influenza transcription and replication but also highlights potential targets for novel antiviral strategies. Overall, this study represents a significant advance in the field.

Minor Comments

1. Figure 1: Consider adding labels beside panel a (“State-In”) and panel b (“State-Out”) to facilitate identification without reading the captions.
2. Figure 1: Please include scale bars for the 2D class averages and reconstructions.
3. Figure 1: Not all NP molecules are labeled with the NP+1, NP-1 system. In particular, for the State-Out panel, labeling all eight NP molecules would enhance clarity.
4. Figure 1: Additionally, in figure 1 caption label the middle panels as isosurface renderings.
5. Supplementary Figure 6: Add “State-In” and “State-Out” labels above the relevant panels, possibly next to the letter labels.
6. Supplementary Figure 6: Label the RNA structures in panels c and d with “5” and “3” to denote the ends of the RNAs.
7. Line 478 in Methods: The word “the” should be capitalized at the beginning of the sentence.

8. The work is very robust, and including supplementary movies with coordinate models and density maps would greatly assist readers. For example:

Supplementary movies for Figure 1a (middle and left panels)

Supplementary movies for Figure 1b (middle and left panels)

Supplementary movies for Supplementary Figure 5 (panels a and b)

This study is a major contribution to the field, and addressing these minor points will further enhance its clarity and impact.

Reviewer #2

(Remarks to the Author)

Influenza virus is a negative-sense, single-stranded RNA virus with a segmented genome. The viral RNA (vRNA) is never found in a naked state but is instead tightly associated with viral proteins—specifically, a single copy of the RNA-dependent RNA polymerase (RdRp) and multiple copies of the nucleoprotein (NP)—to form the viral ribonucleoprotein complex (vRNP), which constitutes the functional replication and transcription unit of the virus. For decades, numerous research groups have attempted to obtain the complete structure of the vRNP. Among early pioneering efforts, a Spanish team

developed a novel strategy to produce a recombinant, synthetic vRNP mimic—termed the “mini-vRNP” (Ortega et al., *J Virol*, 2000; doi: 10.1128/jvi.74.1.156-163.2000). This construct, which shown a circular configuration composed of nine NP monomers and a single RdRp interacting with two of these monomers, provided a foundational tool for structural analyses (Martín-Benito et al., *EMBO Rep*, 2001; doi: 10.1093/embo-reports/kve063, Area et al., *PNAS*, 2004; doi: 10.1073/pnas.0307127101). Despite its artificial nature, this system enabled structural characterization at moderate resolution (15–20 Å), marking a significant step forward in the field.

In the present manuscript, Kang and colleagues adopt a similar approach to generate a mini-vRNP comprising a 248-nucleotides RNA segment. This RNA includes both 5' and 3' terminal sequences that serve as the viral promoter and are bound by a single RdRp complex. The rest of the RNA is encapsidated by multiple NP protomers. Through biochemical purification of mini-vRNPs with varying numbers of NP subunits, the authors performed cryo-electron microscopy (cryo-EM) analysis, achieving their highest-resolution reconstructions with particles containing eight NP monomers. Notably, the authors report two major structural novelties: (1) the polymerase can localize not only on the exterior of the NP ring but also within its interior—a deviation from earlier observations made by the Spanish team; and (2) a single NP protomer looks capable of interacting simultaneously with both the 3' and 5' of the RNA in close association with the RdRp. This latter finding raises intriguing mechanistic questions about whether NP may possess additional roles, such as helicase-like activity or a regulatory “locking” function.

My evaluation proceeded in two phases. In initial readings, I found the structural data describing the NP-0 state—though I note that the nomenclature is potentially confusing due to similarity with “N0P” (the complex between the nucleoprotein (N) and the Phosphoprotein (P) to keep N in a monomeric RNA-free form) from Mononegavirales literature—particularly compelling in terms of the proposed interaction with the polymerase. The “RdRpin” conformation is novel and provides new insights into the conformational flexibility and dynamic architecture of the vRNP. Conversely, the “RdRpout” conformation does not offer significant new information regarding polymerase knowledge, and the structural interpretations of the NP+1–NP+2 appear less convincing. Subsequently, on 15 May, I've been noticed with a recently published study in *Science* describing the structure of an authentic, full-length vRNP (Peng et al., *Science* 2025, doi: 10.1126/science.adq7597). Upon reconsidering Kang and colleagues manuscript in light of this new publication, it gives the feeling that the current submission appears to be a (too?) rapid response to that work. While some data are original, the manuscript lacks the depth and rigor necessary to stand on its own in its present form. It would benefit from a thorough revision that centers exclusively on the novel aspects of the NP–RdRp interaction. In particular, the proposed interaction should be re-examined with reference to recent structural studies published in early 2025 on in vitro reconstituted NP-RNA complexes (Chenavier et al., *NAR* 2025, doi: 10.1093/nar/gkae1211), which was used in the article published in *Science*. Structural alignments made with the PDBs & maps kindly provided by the authors, indicate that residues 71–91 of NP (also predicted also by AlphaFold to form an α -helix when predicting the NP-RNA complex as for the X-ray structure of NP in complex with a nanobody (PDB ID: 5TJW)), might be considered to be an α -helix. Therefore, the current models should be rebuilt, incorporating this conformational analysis (and validation files should be also provided for peer review!!). When refining the NP-RNA interaction model, the authors may additionally find it beneficial to consider analogous work on the Tilapia lake virus (Arragain et al., *NAR* 2025, doi: 10.1093/nar/gkaf112), which may offer additional comparative insights to their model-building strategy.

I believe the data presented in this study have the potential to make a strong impact in the field, comparable to the work published in *Science* by Peng et al. To maximize this impact, the authors are encouraged to refine their manuscript by focusing more tightly on the central message—the organization of the vRNP. Currently, the manuscript attempts to address too many aspects, which dilutes the clarity and strength of the core findings. A more focused presentation would significantly enhance the manuscript's coherence and scientific contribution. In order to re-write/revise the manuscript, the authors should consider the following points:

Major

- The interaction between RdRpin and NP-0 is compelling (please also consider my earlier remark regarding the identification of this NP protomer regarding the context of Mononegavirales N0P complexes). However, what about the conservation of the corresponding sequences? Are these residues strictly conserved across related viruses? Specifically, since R208 of NP interacts with the backbone of residues PA375–377, how conserved are these particular residues in both proteins?
 - The length of the RNA used in the study—248 nucleotides—is not fully addressed in the manuscript. Based on the structural data, the RdRpin–NP-0 complex includes 38 nucleotides, and each NP binds approximately 7–9 nucleotides. This would correspond to roughly $38 + 7 \times 9 = 101$ nucleotides, which is significantly less than the full length RNA. This discrepancy suggests that more than half of the RNA is invisible and so unaccounted for and should be discussed. Furthermore, it appears that the authors assume a direct, linear connection of RNA between individual NP protomers. However, what about a potential continuity of the RNA at the interface between two NP molecules? This alternative should be considered. The strategy used—deriving NP+1–NP+2 maps from a local refinement focused on the RdRpin–NP-0 complex—may not be optimal for addressing this point. Additionally, in Figure Sup3k, the density map of NP-0 appears to be of higher quality than that of NP+1–NP+2, despite the latter having a better nominal resolution. This is unexpected for a flexible complex, where one would anticipate resolution to decrease with increased distance from the refinement center. This inconsistency warrants further clarification.
 - The relevance of the discussion on ANP32A (line 281-319 and Fig 5) to the main focus of the manuscript is not entirely clear. The study primarily investigates the organization of the vRNP using a synthetic system, yet the narrative shifts toward replication, with no direct biological and/or structural data to support this transition (at least is ANP32A interacts with the Mini-RNP?). While the points raised are interesting and highly valuable for the biology of influenza, they feel somewhat disconnected from the core findings. It may be more effective to either streamline this section or consider expanding it in a separate manuscript/review where it can be more fully explored and supported.
- PDBs and Maps. I would like to thank the authors for kindly providing their PDB files and corresponding maps upon request for the review process. Although the official PDB validation reports were not included, I have nonetheless assessed that the

authors have rigorously followed the editorial policies of Nature Communications concerning macromolecular structures (Official validation reports from the wwPDB are required for peer review). Upon examining the PDB files, I noted that the oligomerization loop shares the same chain-ID as the NP core. Could the authors clarify whether they believe RNA binding to NP does not influence oligomerization? Additionally, do the authors consider the possibility that multiple NP molecules may interact with RNA in a non-encapsidated manner? If so, this point should be explicitly discussed in the manuscript. Please also refer to my earlier comments regarding the 71–91 loop. Based on the provided maps, it appears plausible that a loop-to-helix transition in NP may occur upon RNA binding. Such conformational change could be worth considering, especially given that similar transitions can be also induced by nanobody binding alone. If any structural models are revised, I also suggest assigning a distinct chain ID to the RNA molecule(s), different from that of the protein, to enhance clarity.

Minor

- Line 124, Sup3b and Sup4b show 2Dclasses. The 3D reconstructions should be Sup3c&3d and Sup4c instead Sup3b and Sup4b. please confirm
- The Cter is not visible (line251)... what's about the Nter? Make comments
- Line 400-405: difference of magnification and pixel size between the different data collections should be explained.
- The explanation of the triple mutants should be provided earlier in the legend of Figure 2, as it currently appears quite late (lines 747–749). It would be more helpful to include this information in the description of panel 2d, where the mutants are first introduced.
- Please add a scale bar to each micrograph to facilitate interpretation.
- Many figures/panel are not visible: Sup1b (too small! I just sometime see green spots.), Sup3a&4a (maybe add a zoom on the micrograph to be able to distinguish something!), Sup5a&5b (the choice of the colour is not appropriate to show the density and the fitted models, ... too clear compare to the colour of the paper!), Sup8b and Sup11 (too small to see something correctly!).
- Fig Sup11, please label ANP32A!
- If the authors decide to keep the identification NP-0, please homogenize the text and the figures: there is a NP0 (without the -) on Sup3k, NP+0 on Sup10.
- Reference 22: The authors could consider citing a more recent review on single-stranded virus nucleocapsids, as the current one is over 13 years old. Additionally, this particular review may not be the most appropriate source for discussing domain or terminal reorganizations in N/NP. A more targeted research article might be more suitable and informative for supporting that specific point.
- In references, some articles are written with the full length name of the paper, others with abbreviations! Please homogenize.

Version 1:

Reviewer comments:

Reviewer #1

(Remarks to the Author)

The authors did a great job in addressing all the suggested comments. The manuscript is improved, and the results are noteworthy. This work is of significance to the field of structural biology, structural virology and drug development. The work strongly supports the conclusions and claims.

The data analysis is robust and of high quality. The methodology is sound, and the work meets the standard in the field. There is enough detail provided in the methods for the work to be reproduced.

All required reporting summaries and related documents are supplied. This paper is a great advance in the field of the structural biology of replication/transcription for viruses.

Reviewer #2

(Remarks to the Author)

I would like to express my appreciation to the authors for their careful consideration of my comments and the resulting improvements made to the manuscript, which I now consider to be fully suitable for publication. I would also like to commend them on this exemplary piece of work, which significantly advances our understanding of influenza RNP function.

Prior to final publication, however, I wish to highlight a few minor typos that should be addressed:

- Line 234: the sentence requires an initial capital letter.
- Line 301: consider whether "And" is essential in this context at the beginning of the sentence.
- Line 309: it appears that "Howerv" is a misspelling of "However"; please correct accordingly.
- Line 319-320: I would write "in our mini-vRNP structure" or "in our structure of the mini-vRNP" instead of "in our structure of mini-vRNP".

REVIEWER COMMENTS

Reviewer #1 (Remarks to the Author):

The authors present a robust and impactful study on the structure and function of the mini ribonucleoprotein (RNP) complex of the influenza virus. Recombinant molecules of NP, PB1, PB2, PA, and mini vRNA were expressed and used to purify mini-RNPs for structural and biochemical analyses. Techniques such as cryo-electron microscopy, RT-PCR, and mutagenesis RNA system assays were employed to elucidate the structure and function of mini-RNPs.

The study reveals that the mini-RNP complex exists in two major conformational states: State-In and State-Out. This classification is based on the relative position of the viral polymerase in relation to the NP ring. A series of interactions were identified, including NP–NP, NP–viral polymerase, NP–RNA, and viral polymerase–RNA, along with the presentation of viral protein interfaces that interact with cellular proteins. Based on the structural data, the authors propose a model in which molecular interactions and the positioning of the viral polymerase modulate mini-RNP RNA synthesis. This work not only advances our understanding of the structure–function relationship in influenza transcription and replication but also highlights potential targets for novel antiviral strategies. Overall, this study represents a significant advance in the field.

Response: We thank the reviewer for clearly summarizing our main findings and for these positive comments.

Minor Comments

1. Figure 1: Consider adding labels beside panel a (“State-In”) and panel b (“State-Out”) to facilitate identification without reading the captions.

Response: We thank the reviewer for this suggestion. The labels "State-In" and "State-Out" have been added to panels (a) and (b) of Figure 1, respectively.

2. Figure 1: Please include scale bars for the 2D class averages and reconstructions.

Response: As suggested, we have added scale bars to the 2D class averages and reconstructions in Figure 1.

3. Figure 1: Not all NP molecules are labeled with the NP+1, NP-1 system. In particular, for the State-Out panel, labeling all eight NP molecules would enhance clarity.

Response: As suggested, all NP molecules in Figure 1 have been labeled.

4. Figure 1: Additionally, in figure 1 caption label the middle panels as isosurface renderings.

Response: As suggested, we have labeled the middle panels as isosurface rendering in the Figure 1 legend.

5. Supplementary Figure 6: Add “State-In” and “State-Out” labels above the relevant panels, possibly next to the letter labels.

Response: As suggested, we have added “State-In” or “State-Out” labels alongside the letter labels in Supplementary Figure 6.

6. Supplementary Figure 6: Label the RNA structures in panels c and d with “5’” and “3’” to denote the ends of the RNAs.

Response: As suggested, the 5’ and 3’ ends of the RNA structures in Supplementary Figure 6 have been labeled.

7. Line 478 in Methods: The word “the” should be capitalized at the beginning of the sentence.

Response: As suggested, the initial 't' in "the" has been capitalized (line 547).

8. The work is very robust, and including supplementary movies with coordinate models and density maps would greatly assist readers. For example:

Supplementary movies for Figure 1a (middle and left panels)

Supplementary movies for Figure 1b (middle and left panels)

Supplementary movies for Supplementary Figure 5 (panels a and b)

This study is a major contribution to the field, and addressing these minor points will further enhance its clarity and impact.

Response: We thank the reviewer for this valuable suggestion. We have added several movies with coordinate models and density maps in Supplementary movies 1-6.

Reviewer #2 (Remarks to the Author):

Influenza virus is a negative-sense, single-stranded RNA virus with a segmented genome. The viral RNA (vRNA) is never found in a naked state but is instead tightly associated with viral proteins—specifically, a single copy of the RNA-dependent RNA polymerase (RdRp) and multiple copies of the nucleoprotein (NP)—to form the viral

ribonucleoprotein complex (vRNP), which constitutes the functional replication and transcription unit of the virus. For decades, numerous research groups have attempted to obtain the complete structure of the vRNP. Among early pioneering efforts, a Spanish team developed a novel strategy to produce a recombinant, synthetic vRNP mimic—termed the “mini-vRNP” (Ortega et al., *J Virol*, 2000; doi: 10.1128/jvi.74.1.156-163.2000). This construct, which shown a circular configuration composed of nine NP monomers and a single RdRp interacting with two of these monomers, provided a foundational tool for structural analyses (Martín-Benito et al., *EMBO Rep*, 2001; doi: 10.1093/embo-reports/kve063, Area et al., *PNAS*, 2004; doi: 10.1073/pnas.0307127101). Despite its artificial nature, this system enabled structural characterization at moderate resolution (15–20 Å), marking a significant step forward in the field.

In the present manuscript, Kang and colleagues adopt a similar approach to generate a mini-vRNP comprising a 248-nucleotides RNA segment. This RNA includes both 5' and 3' terminal sequences that serve as the viral promoter and are bound by a single RdRp complex. The rest of the RNA is encapsidated by multiple NP protomers. Through biochemical purification of mini-vRNPs with varying numbers of NP subunits, the authors performed cryo-electron microscopy (cryo-EM) analysis, achieving their highest-resolution reconstructions with particles containing eight NP monomers. Notably, the authors report two major structural novelties: (1) the polymerase can localize not only on the exterior of the NP ring but also within its interior—a deviation from earlier observations made by the Spanish team; and (2) a single NP protomer looks capable of interacting simultaneously with both the 3' and 5' of the RNA in close association with the RdRp. This latter finding raises intriguing mechanistic questions about whether NP may possess additional roles, such as helicase-like activity or a regulatory “locking” function.

Response: We are grateful to the reviewer for outlining the background of influenza virus mini-vRNP research and for clearly summarizing our main findings in this area.

My evaluation proceeded in two phases. In initial readings, I found the structural data describing the NP-0 state—though I note that the nomenclature is potentially confusing due to similarity with “NOP” (the complex between the nucleoprotein (N) and the Phosphoprotein (P) to keep N in a monomeric RNA-free form) from Mononegavirales literature—particularly compelling in terms of the proposed interaction with the polymerase. The “RdRpin” conformation is novel and provides new insights into the conformational flexibility and dynamic architecture of the vRNP. Conversely, the “RdRpout” conformation does not offer significant new information regarding

polymerase knowledge, and the structural interpretations of the NP+1–NP+2 appear less convincing. Subsequently, on 15 May, I've been noticed with a recently published study in Science describing the structure of an authentic, full-length vRNP (Peng et al., Science 2025, doi: 10.1126/science. adq7597). Upon reconsidering Kang and colleagues manuscript in light of this new publication, it gives the feeling that the current submission appears to be a (too?) rapid response to that work. While some data are original, the manuscript lacks the depth and rigor necessary to stand on its own in its present form. It would benefit from a thorough revision that centers exclusively on the novel aspects of the NP–RdRp interaction. In particular, the proposed interaction should be re-examined with reference to recent structural studies published in early 2025 on in vitro reconstituted NP-RNA complexes (Chenavier et al., NAR 2025, doi: 10.1093/nar/gkae1211), which was used in the article published in Science. Structural alignments made with the PDBs & maps kindly provided by the authors, indicate that residues 71–91 of NP (also predicted also by AlphaFold to form an α -helix when predicting the NP-RNA complex as for the X-ray structure of NP in complex with a nanobody (PDB ID: 5TJW)), might be considered to be an α -helix. Therefore, the current models should be rebuilt, incorporating this conformational analysis (and validation files should be also provided for peer review!!). When refining the NP-RNA interaction model, the authors may additionally find it beneficial to consider analogous work on the Tilapia lake virus (Arragain et al., NAR 2025, doi: 10.1093/nar/gkaf112), which may offer additional comparative insights to their model-building strategy.

I believe the data presented in this study have the potential to make a strong impact in the field, comparable to the work published in Science by Peng et al. To maximize this impact, the authors are encouraged to refine their manuscript by focusing more tightly on the central message—the organization of the vRNP. Currently, the manuscript attempts to address too many aspects, which dilutes the clarity and strength of the core findings. A more focused presentation would significantly enhance the manuscript's coherence and scientific contribution. In order to re-write/revise the manuscript, the authors should consider the following points:

Response: We sincerely thank the reviewer for the insightful comments on the influenza vRNPs of our manuscript. We wish to clarify that our work of mini-vRNP was completed at last year, which is prior to the Peng *et al.* study (Science 2025, doi: 10.1126/science. Adq 7597). We first encountered this newly published article during the external review process of our manuscript.

Regarding the suggestion to focus the discussion on the novel aspects of RdRp-NP/NP-RNA interactions, we fully concur and have revised the manuscript accordingly. Following the reviewer's advice, we attempted to remodel the residues 72-90 (to be consistent with the published papers, we have standardized the region 71-91 as D72-

K90) of NP protein as an α -helix. During structural refinement, we observed that the density of this region adopted a stable α -helical conformation (residues 72-81; reference PDB entry: 5TJW) in NP-0 (Two validation reports for the revised NP-0 structures have been provided for peer review); while NP+2 exhibited insufficient chain length for helical formation as the electronic density of residues 72-90 appeared more diffuse. Consequently, this region of NP+2 was maintained as a loop in the final model to match its electron density.

Major

- The interaction between RdRpin and NP-0 is compelling (please also consider my earlier remark regarding the identification of this NP protomer regarding the context of Mononegavirales NOP complexes). However, what about the conservation of the corresponding sequences? Are these residues strictly conserved across related viruses? Specifically, since R208 of NP interacts with the backbone of residues PA375–377, how conserved are these particular residues in both proteins?

Response: We thank the reviewer for the constructive suggestion to perform a comparison of PA/NP sequence conservation across related viruses. We have included sequence alignment of NP and PA from representative influenza A viruses in Supplementary Figure 8 to analyze the conservation of these particular residues. The description of the conservation of these key residues has also been added to the text (line 191-194, line 204-206 and line 301-303).

- The length of the RNA used in the study—248 nucleotides—is not fully addressed in the manuscript. Based on the structural data, the RdRpin–NP-0 complex includes 38 nucleotides, and each NP binds approximately 7–9 nucleotides. This would correspond to roughly $38 + 7 \times 9 = 101$ nucleotides, which is significantly less than the full length RNA. This discrepancy suggests that more than half of the RNA is invisible and so unaccounted for and should be discussed. Furthermore, it appears that the authors assume a direct, linear connection of RNA between individual NP protomers. However, what about a potential continuity of the RNA at the interface between two NP molecules? This alternative should be considered. The strategy used—deriving NP+1–NP+2 maps from a local refinement focused on the RdRpin–NP-0 complex—may not be optimal for addressing this point. Additionally, in Figure Sup3k, the density map of NP-0 appears to be of higher quality than that of NP+1–NP+2, despite the latter having a better nominal resolution. This is unexpected for a flexible complex, where one would anticipate resolution to decrease with increased distance from the refinement center. This inconsistency warrants further clarification.

Response: We thank the reviewer for the highly valuable suggestion. Although 248-nt vRNA was used for the mini-vRNP assembly, the resolved structure reveals significantly less than the full length vRNA, we have incorporated speculations concerning this phenomenon into our discussion (line 335-349). We proposed that each NP is capable of binding approximately 24–28 nt of vRNA and the number of nucleobases that binds to NP is not fixed. The nucleotides unresolved in our structures are likely positioned at the interface between two neighboring NPs, these regions have a large open pocket that could accommodate RNA forming stem-loop structures, potentially mediating intersegment RNA interactions. However, significant conformational flexibility prevents the RNA structural determination.

Additionally, we apologise for any confusion that may have existed, in general we believe that the worse density of NP+1/NP+2 is due to it being inherently more flexible, despite having a higher nominal resolution, and this may be due to cryosparc's own algorithms which lead to a degree of overfitting. We also noted the occurrence of overfitting during directly performed local refinement on NP-0, so we performed an additional 3D classification in the region, which did reduce the degree of overfitting. So we also performed the same classification strategy for NP+1/NP+2, which improved the density but we recognise that there is still a degree of overfitting. In fact, we have tried different strategies in reconstructing NP+1/NP+2, such as directly masking NP+1 and NP+2 after global refinement of the mini-RNP or performing local refinement NP+1 and NP+2 respectively, but the process that we have mentioned in the manuscript so far produced the best results. Although the mask may look tight from the FSC curve, it does retain the RNA signal well and allows us to see the density of the NP backbone more clearly.

- The relevance of the discussion on ANP32A (line 281-319 and Fig 5) to the main focus of the manuscript is not entirely clear. The study primarily investigates the organization of the vRNP using a synthetic system, yet the narrative shifts toward replication, with no direct biological and/or structural data to support this transition (at least is ANP32A interacts with the Mini-RNP?). While the points raised are interesting and highly valuable for the biology of influenza, they feel somewhat disconnected from the core findings. It may be more effective to either streamline this section or consider expanding it in a separate manuscript/review where it can be more fully explored and supported.

Response: We appreciate the reviewer's advice to streamline the section of the discussion on ANP32A (line 350-388 and Fig 5). Therefore, following the reviewer's recommendation, in the revised version of the manuscript we have removed the Fig 5

and the final paragraph of the discussion and shifted the section's focus to mini-vRNP conformation.

- PDBs and Maps. I would like to thank the authors for kindly providing their PDB files and corresponding maps upon request for the review process. Although the official PDB validation reports were not included, I have nonetheless assessed that the authors have rigorously followed the editorial policies of Nature Communications concerning macromolecular structures (Official validation reports from the wwPDB are required for peer review). Upon examining the PDB files, I noted that the oligomerization loop shares the same chain-ID as the NP core. Could the authors clarify whether they believe RNA binding to NP does not influence oligomerization? Additionally, do the authors consider the possibility that multiple NP molecules may interact with RNA in a non-encapsidated manner? If so, this point should be explicitly discussed in the manuscript. Please also refer to my earlier comments regarding the 71–91 loop. Based on the provided maps, it appears plausible that a loop-to-helix transition in NP may occur upon RNA binding. Such conformational change could be worth considering, especially given that similar transitions can be also induced by nanobody binding alone. If any structural models are revised, I also suggest assigning a distinct chain ID to the RNA molecule(s), different from that of the protein, to enhance clarity.

Response: We thank the Reviewer for the highly valuable suggestion. During the post-submission period, we re-examined the relevant PDB files and identified that the oligomerization loop shared the same chain ID as the NP core in the coordinate of NP+1:NP+2:RNA unit. We therefore assigned a distinct chain ID to the RNA molecule. We think that RNA binding to NP can influence oligomerization, and consider there is no evidence to suggest that multiple NP molecules may interact with RNA in a non-encapsidated manner.

In addition, following the reviewer's advice, we attempted to remodel the residues 72-90 (to be consistent with the published papers, we have standardized the region 71-91 as D72-K90) of NP protein as an α -helix. During structural refinement, we observed that the density of this region adopted a stable α -helical conformation (residues 72-81; reference PDB entry: 5TJW) in NP-0 (Two validation reports for the revised NP-0 structures have been provided for peer review); while NP+2 exhibited insufficient chain length for helical formation as the electronic density of residues 72-90 appeared more diffuse. Consequently, this region of NP+2 was maintained as a loop in the final model to match its electron density.

Given the distinct conformations of residues 72–90 in NP-0 and NP+2, we propose that NP-0 plays a role in bifurcating the double strand-vRNA and maintaining the vRNA

circularization conformation. The transition from a loop to an α -helix in this region is crucial for stabilizing the vRNA circularization conformation. In contrast, the conformation of residues 72-90 in NP+2 adopts a more open loop positioned above the linear RNA, which enables broader surface coverage and stabilization of the vRNA. The phenomenon also observed in the previously resolved NP and 3-mer RNA co-crystal structure (PDB code: 7DXP). This difference between NP-0 and NP+2 suggests flexible D72-K90 loop is crucial for RNA binding, and flexible NP orientations that may be necessary for native RNP assembly. This viewpoint has been added to the discussion in our manuscript (line 325-331).

Minor

- Line 124, Sup3b and Sup4b show 2D classes. The 3D reconstructions should be Sup3c&3d and Sup4c instead Sup3b and Sup4b. please confirm.

Response: We sincerely thank the reviewer for identifying this error and apologize for the oversight. The original Sup3b and Sup4b (line 126) have been replaced with Sup3c&3d and Sup4c, respectively. Sup3b and Sup4b are now marked following 'these particles' on line 124.

- The Cter is not visible (line251)... what's about the Nter? Make comments.

Response: The Nter is not visible in our structures, too. The absence of visible N-terminal regions in most resolved influenza NP structures suggests inherent conformational flexibility in this domain. As this represents a common feature in the NP conformations, we have not specifically discussed it in the manuscript. It is worth noting that Chenavier et al. (2025) reported *in vitro* reconstitution NP-RNA helical complexes, they obtained the helical assemblies observed looked similar to single-stranded nucleocapsid of unwound vRNPs diluted in low salt conditions by deleting the first 14 residues of the NP N-terminal. They demonstrate that NP's N-terminal 14 residues prevent structural stabilization while increasing helical flexibility. These residues confer vRNP flexibility and enable nuclear trafficking of vRNPs and newly synthesized NPs.

- Line 400-405: difference of magnification and pixel size between the different data collections should be explained.

Response: Due to the difference in direct electron detector, there is a difference in magnification and pixel size available for these two microscopes. Considering the actual size of the particles, we chose the appropriate magnification and pixel size for

the data processing respectively. In fact, we collected the first set of data with the Falcon4 direct electron detector, in which we found that the mini-vRNPs mainly existed in the form of state in, but a small amount of state out mini-vRNPs existed in the sample. Due to the amount of data, we could not get any 3D information of the state out mini-vRNPs. So we continued to collect another set of data to increase the number of particles. The Titan Krios electron microscope equipped with a K3 direct electron detector was available in plenty of time for us to use it, so all subsequent data collection was done on this microscope.

- The explanation of the triple mutants should be provided earlier in the legend of Figure 2, as it currently appears quite late (lines 747–749). It would be more helpful to include this information in the description of panel 2d, where the mutants are first introduced.

Response: We thank the reviewer for this suggestion. As requested, the explanation of the triple mutants has been relocated from the original text (lines 821–822) to the description of panel 2d (lines 817–818).

- Please add a scale bar to each micrograph to facilitate interpretation.

Response: As suggested, we have added scale bars to the 2D class averages and reconstructions in Figure 1, Supplementary Figure 1, Supplementary Figure 3 and Supplementary Figure 4.

- Many figures/panel are not visible: Sup1b (too small! I just sometime see green spots.), Sup3a&4a (maybe add a zoom on the micrograph to be able to distinguish something!), Sup5a&5b (the choice of the colour is not appropriate to show the density and the fitted models, ... too clear compare to the colour of the paper!), Sup8b and Sup11 (too small to see something correctly!).

Response: We thank the reviewer for this suggestion. These figures/panels have now been revised accordingly.

- Fig Sup11, please label ANP32A!

Response: As suggested, ANP32A has been labeled in Supplementary Figure 12 (the original Sup 11).

- If the authors decide to keep the identification NP-0, please homogenize the text and

the figures: there is a NP0 (without the -) on Sup3k, NP+0 on Sup10.

Response: We thank the Reviewer for identifying this terminological inconsistency. The designation "NP-0" has now been standardized throughout the text and figures.

- Reference 22: The authors could consider citing a more recent review on single-stranded virus nucleocapsids, as the current one is over 13 years old. Additionally, this particular review may not be the most appropriate source for discussing domain or terminal reorganizations in N/NP. A more targeted research article might be more suitable and informative for supporting that specific point.

Response: We appreciate the Reviewer's advice to citing more suitable review/articles on single-stranded virus nucleocapsids. Following the Reviewer's recommendation, we have addressed this by replacing Reference 22 with current literature: Reference 26 (comprehensive review, 2023) now supports the viewpoint at line 258, while References 29-30 (targeted research articles) provide targeted evidence at line 284.

- In references, some articles are written with the full length name of the paper, others with abbreviations! Please homogenize.

Response: We thank the Reviewer for identifying the non-standard formatting in our references. All references have now been thoroughly checked and standardized to ensure consistent formatting.

REVIEWER COMMENTS

REVIEWERS' COMMENTS

Reviewer #1 (Remarks to the Author):

The authors did a great job in addressing all the suggested comments. The manuscript is improved, and the results are noteworthy. This work is of significance to the field of structural biology, structural virology and drug development. The work strongly supports the conclusions and claims.

The data analysis is robust and of high quality. The methodology is sound, and the work meets the standard in the field. There is enough detail provided in the methods for the work to be reproduced.

All required reporting summaries and related documents are supplied. This paper is a great advance in the field of the structural biology of replication/transcription for viruses.

Response: We sincerely appreciate the reviewers' positive evaluation of our work and their recognition of its significance.

Reviewer #2 (Remarks to the Author):

I would like to express my appreciation to the authors for their careful consideration of my comments and the resulting improvements made to the manuscript, which I now consider to be fully suitable for publication. I would also like to commend them on this exemplary piece of work, which significantly advances our understanding of influenza RNP function.

Response: We thank the reviewers for acknowledging the importance of our findings and for their thoughtful suggestions that helped improve the manuscript.

Prior to final publication, however, I wish to highlight a few minor typos that should be addressed:

- Line 234: the sentence requires an initial capital letter.

Response: We thank the reviewer for pointing out the typographical error. It has now been corrected throughout the revised manuscript.

- Line 301: consider whether “And” is essential in this context at the beginning of the sentence.

Response: We sincerely thank the reviewer for this thoughtful suggestion. After careful consideration, we have retained this statement, as NP-0 plays dual roles in separating the RNA duplex and stabilizing it. We hope the reviewer agrees that this clarification strengthens the overall description.

- Line 309: it appears that “Howerv” is a misspelling of “However”; please correct accordingly.

Response: We thank the reviewer for pointing out the spelling errors and apologize for the oversight. We have carefully corrected it in the revised manuscript.

- Line 319-320: I would write “in our mini-vRNP structure” or “in our structure of the mini-vRNP ”instead of “in our structure of mini-vRNP”.

Response: We are grateful for the reviewer’s attention to detail. It has been corrected in the revised manuscript.